# Low-Altitude UAV-Based Recognition of Porcine Facial Expressions for Early Health Monitoring

**DOI:** 10.3390/ani15233426

**Published:** 2025-11-27

**Authors:** Zhijiang Wang, Ruxue Mi, Haoyuan Liu, Mengyao Yi, Yanjie Fan, Guangying Hu, Zhenyu Liu

**Affiliations:** 1College of Information Science and Engineering, Shanxi Agricultural University, Taigu 030801, China; 2College of Agricultural Engineering, Shanxi Agricultural University, Taigu 030801, China; 3College of Animal Science, Shanxi Agricultural University, Taigu 030801, China; 4Dryland Farm Machinery Key Technology and Equipment Key Laboratory of Shanxi Province, Taigu 030801, China

**Keywords:** image recognition, YOLO model, aerial inspection, health monitoring, animal welfare

## Abstract

This study targets the critical challenges of pig facial expression recognition in large-scale swine-herd environments and proposes a detection method that couples active-flight inspection with a refined YOLOv8 architecture. The aim is to improve the accuracy and efficiency of automated health monitoring in pigs, focusing on the identification of symptoms associated with common conditions such as cold, fever, and cough. To ensure clinical relevance, the labeling of health-related symptoms was verified using a combination of on-site veterinary assessment and auxiliary tools including infrared thermometry (for fever) and clinical observation protocols. By integrating the Detect_FASFF, PartialConv, and iEMA modules, the improved model attains mean average precision at a 50% IoU threshold (mAP_50_) of 95.5%/95.9%, 95.6%, and 96.4%, respectively, surpassing conventional models by a substantial margin. The results demonstrate that the lightweight, high-precision detector can trigger early disease alerts and support intelligent herd management. This advancement has important implications for improving animal welfare and fostering sustainable livestock production.

## 1. Introduction

Pigs, the largest livestock species worldwide, make monitoring their health status and analyzing expression-driven behavior a matter of great importance for improving production efficiency and ensuring animal welfare [1]. Conventional visual observation is highly subjective and lacks continuity, underscoring the need for real-time monitoring with automated techniques. Facial expressions are key indicators that reflect physiological states such as pain, hunger and disease, and thus constitute a pivotal breakthrough for intelligent surveillance. Existing studies predominantly use classical machine-learning approaches (e.g., support-vector machines, SVMs) or basic deep-learning architectures (e.g., residual networks, ‘ResNet-50’) [2,3]. However, the pig’s facial structure is complex—the contours are irregular rather than regular rectangles, and fur can occlude critical features—while environmental interferences such as lighting and stains are pronounced, leading to low recognition accuracy [4].

Ground inspections at pig farms currently suffer from low efficiency, limited coverage and high labor costs. Aerial inspections, which rely on non-contact image-capture technology, can automatically extract facial features such as pupil dilation state and ear posture, thereby overcoming the limitations of traditional inspections in terms of efficiency, safety and convenience. Their full-coverage, blind-spot-free inspection approach is better suited to locating targets in complex backgrounds. This technology advances precise livestock-condition detection, not only optimizing management processes but also substantially reducing costs.

Notably, Marsort et al. developed a convolution-neural-network-based adaptive pig-face recognition method that achieved 83% accuracy [5]; He Yutong et al. proposed a modified ‘YOLOv3’-based pig-face recognition method with 90.12% accuracy [6]; Hansen et al. introduced a CNN model incorporating convolution, max-pooling and dense-connection modules, thereby improving pig-face recognition performance [7]. Yan Hongwen et al. improved the ‘Tiny-YOLO’ model for pig facial-pose recognition [8]; Psota et al. constructed a fully convolutional neural network for instance segmentation of pigs, obtaining 91% accuracy [9].

A review of prior work reveals a taxonomy spanning identity recognition, pose estimation, and behavioral analysis, with only a limited subset directly targeting health status assessment. While several studies have achieved high performance in identity or segmentation tasks [5,7,9], few have incorporated clinically validated health labels—such as those confirmed by veterinary diagnosis or physiological measurements—to establish a robust link between facial features and specific health conditions. Moreover, recent advances in multimodal behavior analysis, including the application of large language models to interpret livestock behavioral data [10], highlight the potential of integrating diverse data sources for improved health monitoring. This study aims to address this gap by developing an aerial inspection-based framework that utilizes clinically grounded facial expression labels for early health anomaly detection.

Building on these studies, we propose three pig-facial-expression recognition models—‘Detect_FASFF_YOLOv8’, ‘PartialConv_YOLOv8’, and ‘iEMA_YOLOv8’—and conduct comparative experiments with the broader YOLO series to analyze how different facial emotional states map to underlying physical conditions. For example, coughing may be triggered by brief loss of focus, subtle upward rotation or lateral shift in the eye caused by an external dry environment [11]; influenza can manifest as abnormal eye conditions, dry lips or slight tremor when pigs are cold or the weather is hot; and fever may be indicated by facial stiffness or languorous posture arising from intrinsic factors [12]. The association between such expressive signals and health status is increasingly supported by research in animal behavior and precision livestock farming, including studies leveraging computer vision and multimodal data for early anomaly detection [13]. Recognizing these signals allows timely intervention, reducing the risk of disease spread and demonstrating the practical value of pig facial-expression recognition in health management.

Pigs, as mammals with complex emotions, exhibit facial expressions that intuitively reflect physiological states and psychological needs. Accurate recognition of these signals enables researchers to capture group stress responses or discomfort promptly, providing a scientific basis for improving rearing environments and optimizing feeding management [14]. This not only reduces unnecessary animal suffering but also promotes a more humane, sustainable direction for the industry, addressing societal ethical demands for animal protection. Real-time monitoring of expression changes equips farmers with early warnings of health risks, informs adjustments to feeding regimes, and can even aid in structuring groups to minimize aggression, thereby enhancing rearing efficiency and product quality and delivering tangible benefits to agricultural economics.

## 2. Materials and Methods

### 2.1. Research Content and Work

#### 2.1.1. Data Collection Area and Sampling Period

The experimental data for pig facial expression were sourced from a commercial pig-rearing unit located in Pianguan County, Shanxi Province (longitude 110°20′–112°00′ E, latitude 39°16′–39°40′ N, mean altitude 1377 m). The farm is divided into three sections, each measuring 10 m × 7 m, for a total area of 210 m^2^. Data were collected from September 2024 to December 2024, between 09:00 h and 18:00 h.

##### Data-Collection Equipment

An OAK-III-mini3 autonomous, obstacle-avoidance open-source UAV (Beijing AVIC Hengtuo Company, Beijing, China) equipped with a 7 cm baseline stereo-vision sensor (Guangdong Shenzhen Canon Official Flagship Store, Shenzhen, China) was used to generate real-time pig-pen maps, precisely locate individual pigs, and capture frontal facial expressions. A Canon EOS 200D camera (effective pixel count ≈ 24.1 MP) (Guangdong Shenzhen Canon Official Flagship Store, Shenzhen, China) was mounted on the drone to produce high-resolution video segments and the specific configuration for this study was as follows:

Lens and Focal Length: Standard 18–55 mm f/3.5–5.6 kit lens, fixed at 35 mm.

Flight Altitude: 3–4 m above the pen floor.

Camera Angle: Approximately 30° downward tilt from the horizontal axis.

Minimum Face Pixel Size: >150 × 150 pixels per face in the captured imagery.

During the data-collection phase, this study employed a more advanced, safer, and higher-accuracy UAV platform. Compared with the conventional ground-camera and crane-mounted approaches used for pig-group information acquisition [15], the primary advantages of the UAV deployment are: (1) a wider capture range that yields more complete and accurate datasets, ensuring exhaustive coverage of pig facial features; (2) reduced personnel movement, enabling sterile operations in modern intensive production systems; and (3) lowered pig stress, as the non-intrusive approach eliminates the agitation that typically occurs when handlers approach pens during ground- or crane-based collection, thereby preventing disturbance of the animals and improving the reliability of the final data.

To mitigate the potential stress induced by UAV acoustic emissions, this work adopted a composite strategy of “noise-reduction technology + desensitization training + temporal prioritization” [16]. The UAV selected for deployment featured a low-decibel propulsion system and was equipped with a sound-attenuating enclosure, which maintained the sound pressure level at pig level below 70 dB at the operational altitude of 3–4 m. Flight speed and altitude were carefully controlled. Prior to formal data collection, a systematic desensitization protocol was implemented, consisting of 5-day brief, non-invasive UAV exposures (15–20 min per session, twice daily) until no overt avoidance or prolonged startle behaviors (e.g., sudden flight, frantic vocalizations) were observed for two consecutive sessions, and data acquisition was scheduled during periods of relative emotional stability (e.g., feeding times, midday rest). Active avoidance of high-activity windows (early morning, late afternoon) and sensitive physiological phases (parturition, weaning) was implemented. Ambient pen sounds were also leveraged to mask residual UAV noise.

Furthermore, obstacle-avoidance and anti-blur algorithms were employed to plan optimal traversals through the pen, ensuring high-quality visual data while minimizing any additional stress on the animals (see Figure 1). This comprehensive approach enhances the intelligence of herd-management systems and simultaneously upholds animal welfare, representing a critical development direction for future intensive data acquisition in pig production.

For each behavioral state, distinct facial-expression datasets were collected to capture both frontal and lateral views. The UAV recorded 50 video sequences, each lasting 5–7 h with a frame rate of 25 fps. Initially, pig facial feature images were manually extracted and classified by health status. These features were then monitored in real-time within the farm setting to assign the corresponding health labels (e.g., a video showing coughing induced by external, dry conditions, or one depicting fever and cold-related symptoms during heat or chill periods).

Dynamic field conditions pose inherent technical challenges—particularly the agitation of the herd induced by UAV flight and the potential bias introduced by camera-captured data—requiring stringent data-validation procedures. Consequently, any image affected by UAV-induced facial blur, herd disturbance, transient occlusion, or poor illumination was designated as ‘invalid data’ and promptly excluded. As illustrated in Figure 2, this rigorous filtering yielded a final dataset of 2800 labeled images. This collection not only covers a wide array of sample types but also demonstrates high reliability.

#### 2.1.2. Facial-Expression Features in Pig Health States

The health-state labels (normal, cold, cough, fever) were assigned through a rigorous multi-step protocol to ensure clinical ground truth. Initially, a standardized labeling flowchart (see Figure 3) was established, outlining the progression from raw video data to final annotation [17]. The process began with the extraction of video clips, which were then preliminarily annotated by two trained technicians focusing on the facial features detailed below. These preliminary labels were subsequently confirmed by a licensed veterinarian based on a comprehensive assessment that integrated the visual annotations with additional clinical measurements: fever was explicitly validated by rectal thermometry (≥39.5 °C defined as febrile), and cough labels were confirmed through synchronized audio recording analysis and clinical examination to rule out non-pathogenic causes. This protocol resulted in a high inter-rater agreement (Cohen’s Kappa > 0.85) between the technicians and the veterinarian prior to the model training phase.

A review of the literature and field observation yielded systematic characterisations of pig facial appearance under four conditions: normal (healthy), cold, cough, and fever. In the normal state, pigs display bright, alert eyes with no erythema, swelling, or discharge; the eyelids are clear, and the corners of the eyes are free from crust. The nasal mucosa is moist and lustrous (typically retaining a thin film of sweat-water), with no rhinorrhea or crusting. The oral cavity is clean, with no salivary exudate or foreign material; the mouth may be naturally closed or slightly open during respiration, resulting in a relaxed expression. In Cases of Common Cold: Affected pigs typically present with a dry and lackluster snout, which may be accompanied by clear or mucoid nasal discharge. The eyes appear slightly dull, and pigs may frequently keep their mouths partially open to facilitate breathing. During Febrile Episodes (Elevated Body Temperature): The facial demeanor is characterized by fatigue and listlessness [18]. Ocular signs include dullness and possibly sticky discharge at the canthus. The snout is often dry and may become fissured.

In Cases of Coughing: A cough is frequently followed by a swallowing motion. Expulsion of nasal secretions may occur during coughing fits, and the mouth corners may have adherent foam or mucus. During frequent coughing episodes, the eyes often exhibit an expression of irritability or distress, reflecting the associated discomfort. The following Table 1 summarizes the salient facial cues that facilitate rapid health assessment:

These phenotypic descriptors underpin the construction of a labeled facial-expression dataset. Each image was first annotated by specialist farm veterinarians, with additional contextual information gleaned from video-based situational analysis. Facial region descriptors were embedded alongside the raw images to produce a structured, machine-readable dataset.

#### 2.1.3. Dataset Partitioning, Extension and Pre-Processing

The dataset expansion adhered to a rigorous data-engineering pipeline. To prevent data leakage and ensure robust evaluation, the dataset was partitioned at the level of individual animal IDs, guaranteeing that all images (including consecutive video frames) of any single pig were exclusively assigned to either the training, validation, or test set. This approach prevents near-duplicate samples from inflating performance estimates. Furthermore, all data augmentation was applied strictly to the training set post-splitting, ensuring that no augmented variants of an image could cross into the validation or test splits. Detailed as follows: Standardization

All raw images were resized to a common resolution of 640 × 640 pixels, ensuring spatial uniformity across the cohort.

2.Noise Mitigation

A denoising routine based on the discrete wavelet transform was applied to suppress high-frequency artifacts. This was followed by Z-score normalization. scaling the pixel intensity distributions across the entire dataset to a mean of 0 and a standard deviation of 1. This two-step preprocessing pipeline established a clean and consistent baseline prior to data augmentation, ensuring that subsequent models were trained on comparable and artifact-minimized data [19,20].

3.Multi-modal Augmentation

All augmentations were applied on-the-fly during model training to maximize data diversity while conserving disk space. This approach ensures that for each of the 100 training epochs, the model encounters a uniquely perturbed version of every training image, thereby significantly improving generalization. The label-preserving transformations, implemented with the following parameter ranges, were concatenated to generate a diversified sample set:

Geometric: Random rotation (±15°), horizontal flip (50% probability).

Photometric: Color jitter (brightness, contrast, and saturation adjustment factors within ±20%).

This strategy bolstered the model’s robustness against pose variations and environmental perturbations [21].

4.Dataset Partitioning

To ensure a representative and unbiased evaluation, the final augmented dataset of 5600 images was partitioned into training, validation, and test sets using a stratified random split with a fixed random seed of 42. This strategy preserved the original class distribution across all splits. The dataset was divided into 3800 images for training, 1100 for validation, and 700 for testing. The precise per-class image counts for each split are detailed in Table 2 below.

This layered optimization framework guarantees high-quality, statistically homogenous image data while dramatically enhancing sample diversity, thereby furnishing deep-learning models with the consistency and generalization needed for reliable pig-health monitoring.

### 2.2. Training Environment Configuration

The training was performed on a workstation equipped with an Intel^®^ Core™ i9-1255U processor, an NVIDIA^®^ GeForce RTX™ 4060 8 GB GPU, and 32 GB DDR5 RAM. The software environment utilized CUDA 11.8 and cuDNN 8.6.0 for GPU acceleration, with deep learning development conducted in Python 3.8. The primary libraries included PyTorch 1.13.0, OpenCV 4.4.0, and other supporting utilities (e.g., numpy 1.23.0, tqdm 4.67.1).

The model was trained with an input image size of 640 × 640 pixels. The detailed hyperparameters and training schedule were configured as follows:

Optimizer: AdamW, with a weight decay of 0.05.

Learning Rate Schedule: A linear warm-up was applied over the first 3 epochs, followed by a cosine annealing schedule, decaying from an initial learning rate of 1 × 10^−3^ to 1 × 10^−5^.

Other Regularization: Gradient clipping was set to a norm of 1.0.

Early Stopping: Training was halted if the validation mean Average Precision (mAP) did not improve for 15 consecutive epochs to prevent overfitting.

Epochs 100

Batch size 16

The total training time for a single run was approximately 4.5 h. To ensure statistical robustness, the experiment was repeated with three different random seeds (42, 3407, 2025). The model weights from the epoch achieving the highest validation mAP were selected as the final model for each run. Under this criterion, the performance on the test set across all runs was 95.7 ± 0.4% mAP_50_ (mean ± standard deviation).

This configuration ensured sufficient computational throughput for the 5 h video streams and the resulting 5600 image dataset, while maintaining reproducibility across different research groups.

### 2.3. Model Overview

#### 2.3.1. YOLOv8 Model

YOLOv8 offers five variants n, s, m, l, and x that differ in network depth and width. To balance detection accuracy with real-time performance for pig-facial-expression recognition, this work adopts the YOLOv8s configuration, which is well-matched to the computational resources available on the experimental platform [22]. The model follows a classic four-stage architecture, in which each module works in concert to maintain high detection accuracy while minimizing computational complexity, as illustrated in Figure 4.

Our proposed enhancements focus on the detection head: the original structure is replaced with our refined heads incorporating the Detect_FASFF, PartialConv, and iEMA modules. This concerted design aims to enhance multi-scale feature fusion, spatial feature representation, and temporal consistency, thereby maintaining high detection accuracy while minimizing computational complexity.

#### 2.3.2. Detect_FASFF_YOLOv8

This study was conducted in a commercial pig farm in Pianguan, Shanxi Province, employing an optimized Detect_FASFF_YOLOv8 model to classify four distinct facial expressions associated with unique body states in pigs. The methodology comprises the following key innovations:Data Acquisition and Preprocessing:

Aerial video footage was collected using drones, while ground-level images were captured via calibrated DSLR cameras. Frame extraction and manual annotation were performed to create a dedicated dataset of high-quality pig facial images under varying expressions.

2.Model Architecture Improvements:

Adaptive Spatial Feature Fusion (ASFF) for YOLOv8;

Detect_ASFF was developed by modifying YOLOv8s detection head to achieve;

Scale-Invariant Learning: Dynamic feature map weighting across stages.

Double-Stage Refinement: A redesigned three-way detection head to mitigate rapid scale transitions (e.g., piglets × adults).

FASFFHead (Feature-Adaptive Spatial Feature Fusion Head):

A novel four-head variant of YOLOv8s detection head to simultaneously enhance performance on:

Micro-Expressions: Subtle muscle movements (e.g., eye wrinkles).

Large-Area Feedback: Jaw/tongue occlusion during distress vocalizations.

Co-Attention Mechanism:

Context-aware spatial-channel weighting enabled by gradient-adaptive attention maps (e.g., prioritizing the periorbital region during sternal recumbency detection).

The baseline YOLOv8 detection head exhibits certain limitations in achieving precise object identification: when integrating multi-scale features (e.g., upsampling P5 and concatenating with P4), spatial misalignment occurs due to resolution discrepancies; meanwhile, the use of a fixed receptive field for objects of varying sizes leads to the dilution of features representing small targets [23]. These issues often result in false positives and missed detections when applied to precise pig face recognition. To address these challenges, this study introduces a redesigned FASFFHead through secondary innovation based on the ASFF detection head in the YOLOv8 model. The core improvements include precise multi-scale feature alignment, dynamic feature fusion, and attention enhancement, effectively mitigating feature misalignment and insufficient scale adaptation inherent in conventional detection heads. The structure is illustrated in Figure 5.

The proposed improvements in this study focus on the FASFF (Feature Adaptive Spatial Fusion) detection head, with key modifications applied to the Neck–Head interface, the internal architecture of the detection head, and the mechanism for adaptive spatial weighting. In the conventional YOLOv8 framework, multi-scale feature fusion is achieved through a combination of upsampling and top-down pathways, enabling the integration of deep semantic information with shallow spatial details. This is further refined via a bottom-up propagation path. The process ultimately generates three-scale feature maps—denoted in this work as P3, P4, and P5—which are directly passed to their respective detection heads.

The main enhancement introduced in our approach is the incorporation of the FASFF module between the output of the Neck and the input to the detection heads. This module adaptively fuses the P3, P4, and P5 feature maps in a spatially conscious manner, yielding refined and enhanced representations referred to as Fused_P3, Fused_P4, and Fused_P5. Acting as an intelligent feature fusion filter, the FASFF module performs secondary optimization on the outputs of the Path Aggregation Feature Pyramid Network (PAFPN) without modifying its internal structure. With the integration of the FASFF module, the overall detection pipeline of YOLOv8 is updated as follows:

The Backbone extracts hierarchical features from the input image. The PAFPN Neck performs initial multi-scale feature aggregation and outputs P3, P4, and P5. The FASFF module applies adaptive spatial fusion to these feature maps, producing more discriminative fused features: Fused_P3, Fused_P4, and Fused_P5. The Detection Head leverages the fused features to perform the final object classification and localization.

The FASFF module operates through a two-stage process: spatial alignment and co-attention fusion. Let(1)$$P_{l}\in R^{H_{i}\times W_{i}\times C}$$
denote an input feature map at level $l\in \left\{3,4,5\right\}$.

Spatial Alignment:

First, all feature maps are resized to a common spatial resolution via $\left(H_{3}\times W_{3}\right)$ bilinear interpolation or strided convolution, yielding ${\tilde{P}}_{3},{\tilde{P}}_{4},{\tilde{P}}_{5}$

2.Co-attention Fusion:

For each spatial location $\left(i,j\right)$, the module generates a set of adaptive attention weights that govern the fusion of features from different levels. This is implemented via a lightweight convolutional sub-network, $F_{attn}$, which takes the concatenated features as input:(2)$$W=F_{attn}([{\tilde{P}}_{3},{\tilde{P}}_{4},{\tilde{P}}_{5}])$$
where $W\in R^{H_{3}\times W_{3}\times 3}$ contains the 3-channel weight map. A SoftMax function is applied along the channel dimension to normalize the weights $\alpha_{i,j}^{l}$ for each level l and location $\left(i,j\right)$, ensuring $\sum_{l}\alpha_{i,j}^{l}=1$.

The final fused feature map F is computed as a weighted sum:(3)$$F_{i,j}=\alpha_{i,j}^{3}\cdot {\bar{P}}_{3,i,j}+\alpha_{i,j}^{4}i,j\cdot {\bar{P}}_{4,i,j}+\alpha_{i,j}^{5}i,j\cdot {\bar{P}}_{5,i,j}$$

F is then split and resized (if necessary) to produce the final enhanced features Fused_P3, Fused_P4, and Fused_P5 for the detection heads [24].

To quantitatively evaluate the contribution of each sub-component within the FASFF module, we conducted a comprehensive ablation study. The baseline is the standard YOLOv8s model. The results, measured by mean Average Precision (mAP_50_) and computational complexity (FLOPs), are summarized in Table 3.

Analysis:

Naive Fusion (element-wise addition of aligned features) provides gain (+11.7% mAP), demonstrating the basic benefit of feature integration.

Spatial Alignment alone significantly improves performance (+12.4% mAP), underscoring its importance in resolving scale discrepancies.

The complete FASFF module with co-attention mechanism yields the highest improvement (+15.4% mAP), validating that adaptive, spatially variant weighting is crucial for optimal multi-scale feature fusion.

In this study, we propose a novel FASFFHead detection head by refining the original YOLOv8 detection head from four key aspects. By integrating intelligent fusion, multi-scale coverage, efficient feature propagation, and a lightweight design, the proposed head effectively mitigates the inherent limitations of conventional single-stage detectors. As demonstrated in Figure 6 and Figure 7, it achieves superior performance in detecting small objects, more stable localization, enhanced robustness, and controlled computational overhead. This detection head not only improves the operational efficiency of YOLOv8 but also strengthens its ability to capture and represent complex semantic information in images, underscoring its significant applicability in object detection—including consistent performance across both large and small targets.

#### 2.3.3. PartialConv_YOLOv8

The conventional YOLOv8 model often encounters high computational complexity, significant memory access overhead, and slow inference speeds when processing large-scale recognition tasks, making it increasingly difficult to meet the essential real-time and accuracy demands in modern livestock farming [25]. To better serve the pig farming industry and achieve rapid and accurate identification of pig facial expressions—enabling timely health assessment by farm personnel—this study focuses on constructing a more efficient neural network. A re-examination of commonly used operators reveals that low Floating Point Operations (FLOPs) do not necessarily translate into high speed, as the bottleneck often lies in frequent memory access, particularly in depthwise convolution operations. Inspired by this insight, we propose a lightweight design strategy based on Partial Convolution (PConv) and introduce a novel module called CSPPC to replace the C2f module in the original YOLOv8 architecture. The proposed CSPPC is a convolution structure engineered for high-speed inference, significantly enhancing network computational efficiency. By leveraging PConv to process redundant information in feature maps, it effectively reduces both computational cost and memory access requirements. As illustrated in Figure 8.

In standard convolution, the convolutional kernels for each output channel are fully connected to all input channels—a configuration often termed “dense connectivity.” This design leads to a quadratic increase in both parameter count and computational cost as the number of channels grows. To address this issue, the Partial Convolution (PConv) method adopted in this study breaks away from the fully connected paradigm by performing convolution only on a subset of input channels, thereby establishing ‘sparse connectivity’. This approach significantly reduces parameters and computational load.

PConv introduces a binary selection matrix, with elements set to 0 or 1, to determine which input channels participate in the convolution operation—where 1 indicates a selected channel and 0 indicates an omitted one. This sparse connectivity not only lowers computational complexity but also decreases memory access, ultimately accelerating neural network inference.

The core innovation lies in partitioning the input channels into multiple groups (typically two), where each convolutional kernel connects only to a subset of input channels rather than all of them. For example, given C input channels, PConv divides them into C_1_ and C_2_ (with C_1_ + C_2_ = C). Two corresponding sub-kernels then process C_1_ and C_2_ separately, and their outputs are merged via concatenation or element-wise addition. This grouping strategy enhances model sparsity, enabling each sub-kernel to focus on different feature subsets and helping extract more diversified representations. Moreover, by adjusting the number and size of groups, the model’s complexity and generalization capability can be flexibly controlled [26].

This design not only optimizes computational resource utilization but also provides greater flexibility for subsequent lightweight adaptations. As a result, the enhanced YOLOv8 model exhibits higher efficiency and accuracy in the task of pig-specific facial expression recognition. As illustrated in Table 4.

By integrating the CSPPC structure, the conventional YOLOv8 model leverages the lightweight properties of Partial Convolution to significantly reduce both parameter count and computational complexity while maintaining detection accuracy. Through a split-merge strategy, partial channel convolution, and integration with PWConv, this study achieves an effective balance between efficient feature extraction and resource consumption, offering a novel optimization solution for pig facial expression detection. The corresponding evaluation results of this module are presented in Figure 9 and Figure 10**.**

#### 2.3.4. The iEMA_Attention Mechanism

To clarify the nomenclature, we explicitly distinguish between the Efficient Multi-scale Attention (EMA) mechanism used in our architecture and the unrelated concept of Exponential Moving Average (EMA) employed for model weight averaging. Our proposed module, termed iEMA_Attention, integrates the EMA attention mechanism with an inverted Residual Mobile Block (iRMB) structure. This lightweight feature enhancement module is designed to improve detection accuracy while maintaining the real-time performance of the baseline YOLOv8 framework, particularly addressing challenges posed by unstable illumination and drone motion jitter in porcine facial feature extraction [27]. The proposed iEMA is particularly suited for scenarios characterized by small targets and complex backgrounds.

The iRMB component is integrated within this module, serving as the fundamental processing unit that enables efficient feature transformation and refinement.

##### Efficient Multi-Scale Attention (EMA)

The EMA mechanism addresses limitations of traditional attention modules like Coordinate Attention (CA) by incorporating parallel convolutional pathways. While conventional approaches rely solely on 1 × 1 convolutions, limiting their ability to capture multi-scale spatial information, the EMA mechanism introduces a parallel branch with 3 × 3 convolutional kernels alongside the standard 1 × 1 branch. This parallel architecture enables the network to simultaneously capture both local fine details and broader contextual information through differently sized receptive fields. The outputs from these parallel branches are fused using a matrix dot-product operation, enhancing feature interaction while maintaining computational efficiency.

##### Inverted Residual Mobile Block (iRMB)

The iRMB component provides an efficient computational foundation for the attention-enhanced features. Employing an inverted residual structure with depthwise convolutions, it enables effective feature transformation with minimal computational overhead. Within our iEMA-Attention module, the iRMB operates on the attention-refined features to further enhance its discriminative capability while maintaining parameter efficiency [28].

##### The Secondary Integration Innovation of iEMA_Attention Mechanism

Facial expression recognition in pigs should not rely solely on visual information but can be enhanced by incorporating multimodal data such as vocalizations and behavioral patterns. For instance, pigs may emit distinct vocal cues when experiencing pain or excitement, and their activity behaviors may correlate with specific facial expressions. Solely employing the EMA mechanism to process image features would fail to fully leverage these multimodal signals, thereby limiting the model’s comprehensive understanding of porcine affective states. Furthermore, given the diversity and dynamism of pig expressions, relying exclusively on the EMA mechanism for static image analysis may prove inadequate for accurately recognizing dynamically changing expressions in real-world scenarios.

The relationship between local features (e.g., subtle changes in eyes or snout) and global features (e.g., the overall expressive context of the face) is particularly important. The EMA mechanism alone may not sufficiently explore such interdependencies, potentially leading to misinterpretations of complex expressions. In pig housing environments with significant variations, such as areas with extreme brightness or darkness, the mechanism may struggle with effective identification and tracking. Additionally, it would be incapable of recognizing rare expressions not represented in the training dataset, being constrained to classifying only those expressions encountered during training.

Compared to the EMA mechanism, the iRMB structure offers distinct advantages: it captures subtle expression variations more effectively, reduces misclassification rates, and—more importantly—its lightweight design minimizes computational overhead. It also demonstrates robustness under challenging lighting conditions and adapts well to different pig breeds and individual variations. Based on these considerations, this study integrates the iRMB module with the EMA module, forming a hybrid architecture as illustrated in Figure 11.

The iRMB mechanism, with its multi-branch parallel processing architecture, enables the extraction of local features from porcine facial expressions across different scales and dimensions. Subtle characteristics such as facial morphology and the degree of eye opening can be effectively captured through this design. In contrast, the EMA mechanism excels at capturing global contextual information. By preserving cross-channel features while reducing computational costs, it enhances the model’s comprehension of overall facial expression patterns in pigs. For instance, when recognizing fear expressions in pigs, the iRMB module can extract local features such as ears pulled backward against the head and widened eyes, while the EMA mechanism captures global characteristics like overall facial muscle tension and tightened facial contours. The integration of these two mechanisms allows for a more comprehensive and precise extraction of porcine facial expression features, leveraging their complementary strengths in local detail representation and global context modeling.

##### Ablation Study

We conducted a comprehensive ablation study to evaluate the individual contributions of the EMA and iRMB components. The baseline configuration was the YOLOv8s model enhanced with our FASFF module. The results, summarized in Table 5, demonstrate the progressive improvements achieved by each component.

The results indicate that the EMA component alone provides substantial improvement (+0.6% mAP), demonstrating its effectiveness in capturing multi-scale contextual information. The iRMB component contributes a moderate gain (+0.3% mAP), highlighting its role in feature refinement. The complete iEMA_Attention module achieves the highest performance (+0.9% mAP), confirming the synergistic effect between the multi-scale attention mechanism and the efficient feature transformation capability of the iRMB, with only a marginal increase in parameters.

Fluctuations in illumination, caused by factors such as time and weather in pig houses, lead to significant variations in the brightness and contrast of facial images [29]. The integrated model compensates for these challenges through iRMB’s multi-scale feature extraction and EMA’s capacity for capturing global context, allowing it to learn stable, illumination-invariant features. The final identification results are provided in Figure 12 and Figure 13.

## 3. Results

### Performance Comparison with Other YOLO Series Models

Assessing whether an improved model achieves its intended objectives is crucial in the highly complex task of porcine facial expression detection and recognition. This study employs comparative experiments to validate the model’s effectiveness, aiming to directly demonstrate its superiority and stability. Our focus extends beyond achieving high quantitative metrics during training; it is equally placed on ensuring the model’s stability and reliability in practical applications, particularly in the challenging scenario of identifying stress-induced expressions. This context involves subtle facial cues and variable environmental conditions.

In our comparative experiments, all models were trained from scratch without using pre-trained weights.

(1) The traditional YOLOv8 model was compared with various improved versions using metrics including mAP@0.5, mAP@0.5:0.95, recall, and loss function. Substantial improvements across multiple metrics were observed, highlighting the model’s potential value for health monitoring in intelligent pig farming environments. Timely and accurate disease identification is critical for preventing large-scale outbreaks. Furthermore, the lightweight design of our model makes it particularly suitable for practical deployment on pig farms. The results are shown in Figure 14 below.

(2) The improved YOLOv8 model was compared against established models including YOLOv5s, YOLOv8s, YOLOv11s, YOLOv12s, YOLOv13s, Faster R_CNN, and RT_DETR. A multi-metric comparative analysis fully demonstrates the significant advantage of the proposed improvements for intelligent porcine health monitoring. The results are presented in Table 6 below.

(3) A comparative analysis of the loss functions between the three improved models and the baseline YOLOv8 is presented in Figure 14d.

$L_{box}$ is the bounding box regression loss, calculated using Distribution Focal Loss (DFL) and Complete Intersection over Union (CIoU). DFL optimizes the probability distribution of the bounding box coordinates around the ground truth, leading to more precise localization, especially for small targets like pig faces. CIoU considers the overlap area, center point distance, and aspect ratio consistency, providing a superior geometric measure for bounding box alignment compared to standard IoU.

$L_{cls}$ is the classification loss, computed via Binary Cross-Entropy (BCE) with label smoothing. This component is responsible for penalizing incorrect health state predictions (Normal, Cold, Cough, Fever). Label smoothing mitigates overfitting by preventing the model from becoming overconfident in its predictions.

$L_{dfl}$ is the Distribution Focal Loss component, which specifically refines the bounding box localization by focusing on learning a more accurate distribution of the box coordinates relative to the ground truth.

This multi-component loss function ensures a balanced optimization towards both accurate spatial localization of the pig’s face and precise classification of its health state, which is critical for reliable health monitoring.

(4) To ensure statistical robustness and mitigate the potential noisiness of single-run evaluations, all performance metrics reported in this study represent the mean values derived from three independent training runs conducted with distinct random seeds (42, 3407, 2025). The variability and stability of the results are indicated by 95% confidence intervals, calculated using the t-distribution to account for the small sample size of runs. As illustrated in Figure 15, these confidence intervals are visually represented as error bars in bar charts and confidence bands in training curves, providing a comprehensive view of model performance consistency across multiple experimental repetitions. This approach ensures that the reported improvements are statistically reliable and not attributable to random variations in training dynamics.

To provide a comprehensive evaluation of model performance across different health states, we present the detailed per-class Average Precision (AP) for each proposed architecture in Table 7. This analysis allows for the identification of class-specific strengths and weaknesses, particularly in distinguishing between clinically similar conditions.

Analysis of Class-wise Performance Patterns:

All three architectures demonstrated highly robust performance in identifying Normal states (AP > 97.0%) and Cough (AP > 99.0%), indicating excellent feature extraction for these clinically distinct conditions. The PartialConv_YOLOv8 model achieved the highest single AP score for the Normal class (98.6%), while the iEMA-YOLOv8 model registered the highest performance for Cough detection (99.4%), establishing it as the most balanced model overall.

The confusion matrix analysis revealed a consistent diagnostic challenge across all models: the highest inter-class confusion persistently occurred between Cold and Fever. Specifically, approximately 4–5% of Cold samples were misclassified as Fever, and 3–4% of Fever samples were misclassified as Cold. This systematic confusion is clinically plausible, as it likely stems from overlapping symptomatology in the early stages of both conditions, particularly non-specific manifestations such as lethargy, ocular changes, and reduced activity levels. Notably, the Detect_FASFF_YOLOv8 architecture, with its dual values representing results from two independent runs, showed consistent and superior performance for Cold detection (90.8%/90.7%) compared to other models, suggesting its multi-scale feature fusion capability provides enhanced discriminative power for the subtle facial features associated with early cold symptoms. Meanwhile, the PartialConv_YOLOv8 model demonstrated the highest AP for Fever detection (95.2%). The closely aligned dual values for the Detect_FASFF model further underscore its reliable and stable performance.

These per-class AP values and confusion patterns provide critical insights for future model refinement, highlighting the specific need for enhanced feature representation to better discriminate between physiologically similar conditions, particularly through the incorporation of temporal symptom progression or additional physiological correlates.

## 4. Discussion

(1) Following a systematic analysis and field investigation of modern intensive pig farms, this study employed aerial inspection using drones for experimental data collection. This approach was selected due to its unparalleled advantages over traditional ground inspection in enhancing biosecurity levels, enabling efficient macroscopic monitoring, and strengthening security and environmental oversight. It is important to clarify that this decision does not aim to completely replace necessary ground inspections but rather to restructure the inspection framework using technological means to achieve risk isolation and a significant boost in operational efficiency.

Traditional farm management heavily relies on personnel entering production areas for ground inspections. However, with the continuous expansion of farming scale and the persistent threat of major diseases such as African Swine Fever (ASF), the limitations of this conventional model have become increasingly apparent [29]. Inspection personnel themselves represent a significant biosecurity risk, as their frequent movement across zones increases the probability of disease transmission. Furthermore, the traditional model suffers from insufficient capability for macroscopic surveillance of large farms, making it difficult to quickly and comprehensively assess the overall status of the operation. Inspection efficiency is particularly low at critical points like farm perimeters and environmental protection areas, with numerous blind spots.

The emergence of aerial inspection technology (drones) offers a revolutionary solution to these challenges. Based on comprehensive data collection and verification, data from a gestation sow farm serves as an example: manual inspection could only cover 60% of the pens, whereas a drone, cruising at 2 m/s with a single 30 min flight, could cover 10 standard barns, increasing the coverage rate to 98%. While ground inspection relies on manual problem identification, taking an average of 4 h to detect anomalies, the drone, operating on pre-set routes twice daily (08:00/15:00) and integrated with the improved model from this study for analysis, can locate problematic pigs within 20 min. The system effectively flags individuals suspected of fever, which, as defined in Section 2, was validated by a rectal temperature exceeding 39.0 °C. The implementation of this technology at the farm successfully prevented two potential Porcine Reproductive and Respiratory Syndrome (PRRS) outbreaks. Additionally, The emergence of aerial inspection technology (drones) offers a promising solution to these challenges. In a representative application at a gestation sow farm, the drone system demonstrated substantial improvements over conventional ground-based methods. It achieved near-complete pen coverage through automated flights, significantly reduced the time required to identify pigs showing potential health anomalies, and facilitated timely intervention. Furthermore, the system employed a structured digital workflow, utilizing wireless data transmission and edge computing to securely synchronize and back up environmental and operational data. This integrated approach enhanced both the efficiency of herd monitoring and the reliability of data management on the farm.

(2) The selection of baseline models for comparison was conducted under a consistent experimental protocol, wherein all models—including YOLOv8s, Faster R_CNN, RT_DETR, YOLOv11s, YOLOv12s and YOLOv13s—were trained from scratch on our pig facial expression dataset to ensure a fair and controlled evaluation. This approach, while eliminating the potential bias introduced by pre-trained weights and ensuring equal adaptation to the specific characteristics of porcine facial features, carries an inherent limitation. It may not fully reflect the performance that architectures like Faster R_CNN or RT_DETR could achieve with large-scale pre-training, which is often leveraged in practice to boost convergence and final accuracy, particularly when target task data is limited. Within this framework, the advantages of our YOLOv8-based architecture are pronounced [30]. Faster R_CNN, as a two-stage detector, utilizes a Region Proposal Network (RPN) to generate candidate regions before performing refinement and classification. While this detailed process can yield high detection accuracy, especially for complex scenes, it incurs a substantially higher computational cost, making it less suitable for the real-time requirements of our aerial inspection application.

Compared to the RT_DETR model, YOLOv8 benefits from an exceptionally robust and mature ecosystem. It maintains competitive accuracy while featuring a very compact model size, low computational complexity (FLOPs), and minimal memory footprint. Even the smallest RT_DETR model, due to the inherent nature of its Transformer architecture, typically exhibits higher parameter counts and computational requirements than a comparable YOLOv8 model.

Compared to YOLOv5, YOLOv8 represents the State-of-the-Art (SOTA) in the YOLO series, building upon the successes of its predecessors. YOLOv8 introduces new features and improvements designed to enhance performance and flexibility. Notable innovations include a redesigned backbone network, a novel anchor-free detection head, and a new loss function. While YOLOv11, YOLOv12 and YOLOv13 may surpass YOLOv8 in certain aspects like precision, their status as newly developed models means their maturity, ecosystem support, and maintenance are still under evaluation. Their reproducible performance across diverse environments and potential unknown vulnerabilities require further validation over time.

Through continuous comparison and in-depth study, YOLOv8 has been identified as an algorithm capable of not only rapid detection but also offering substantial room for improvement, thereby laying the foundation for precisely tailored, application-specific development [31].

## 5. Conclusions

Within the complex emotional repertoire of livestock, facial expressions serve as crucial indicators of affective states. These expressions provide valuable insights into an animal’s emotional well-being and psychological health, rendering their precise detection and classification particularly significant for welfare assessment. Porcine facial morphology presents unique challenges—their muscular architecture is inherently complex, and expressive changes are often subtle and easily overlooked by human observers. To address this gap, we established a comprehensive, multi-scenario dataset of porcine facial expressions, encompassing diverse housing environments and validated emotional states. This richly annotated dataset provides a robust foundation for addressing the complexities of porcine expression recognition.

Leveraging this dataset, we developed three enhanced architectures based on YOLOv8: Detect_FASFF_YOLOv8, PartialConv_YOLOv8, and iEMA_YOLOv8, specifically engineered for accurate classification and identification of porcine facial expressions.

(1) Our improved models demonstrated substantial gains in recognition performance. Detect_FASFF_YOLOv8 incorporates adaptive multi-scale feature fusion through an ASFF mechanism, enabling more intelligent integration of hierarchical feature information and significantly enhancing detection capability across varying target scales. PartialConv_YOLOv8 adopts a lightweight, mask-aware convolutional strategy that confers remarkable robustness against partial occlusions and image imperfections. iEMA_YOLOv8 enhances the attention mechanism via a refined exponential moving average approach that stabilizes training dynamics, improves generalization capacity, and consequently achieves superior accuracy. The integration of these complementary technologies not only elevates computer vision performance but also advances livestock welfare research. To further strengthen the clinical relevance of this system, a future clinical validation plan will be implemented. This will involve a longitudinal study with synchronized collection of video data and gold-standard physiological measurements (e.g., rectal thermometry, veterinary diagnosis) to establish a definitive correlation between algorithm outputs and confirmed health states. This translational research direction holds significant potential for advancing intelligent livestock management practices [31].

(2) Comparative evaluation against established object detection benchmarks, clearly demonstrates the superiority of our enhanced YOLOv8 frameworks. While conventional YOLOv8 sets a commendable standard for inference speed and accuracy, our models consistently exceed this baseline. Experimental results show mAP@0.5 values of 95.5% (small targets), 95.9% (large targets), 95.6%, and 96.4% across different architectural improvements. This advantage becomes more pronounced when comparing against YOLOv5s and contemporary detectors including Faster R_CNN, RT_DETR, YOLOv11s, and YOLOv12s. Across all key metrics, our three proposed variants demonstrate significant improvements, with particular excellence in small target detection and occlusion robustness, thereby validating the efficacy of our architectural innovations.

(3) In direct comparison with YOLOv8s, which achieved a baseline mAP@0.5 of 82.1%, our improved models maintain a substantial advantage of 13.7 percentage points. This performance differential not only confirms consistent outperformance against established benchmarks but also underscores the meaningful progress achieved through our methodological refinements, particularly in balancing accuracy with computational efficiency.

Despite these promising results, several limitations warrant attention. Our dataset, though diverse in environmental conditions, remains susceptible to confounding factors that occasionally limit optimal recognition accuracy. Furthermore, aerial data collection presents operational challenges including potential animal disturbance and connectivity issues. Future work will focus on methodological refinements and dataset expansion to incorporate broader scenarios and genetic varieties. These efforts will enhance model robustness and generalizability, facilitating practical implementation in real-world livestock management systems and ultimately contributing to enhanced animal welfare standards.

## Figures and Tables

**Figure 1 animals-15-03426-f001:**
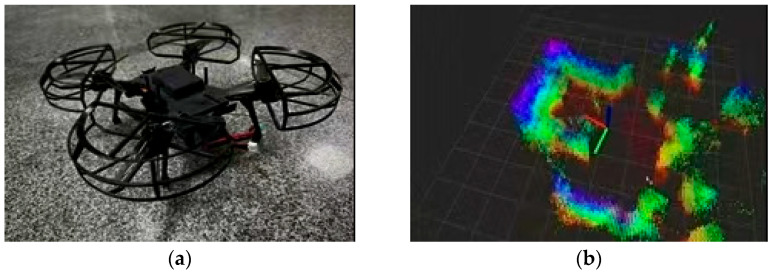
(**a**) OAK-III mini3 autonomous obstacle-avoiding UAV; (**b**) Cloud-based flight-path planning map.

**Figure 2 animals-15-03426-f002:**
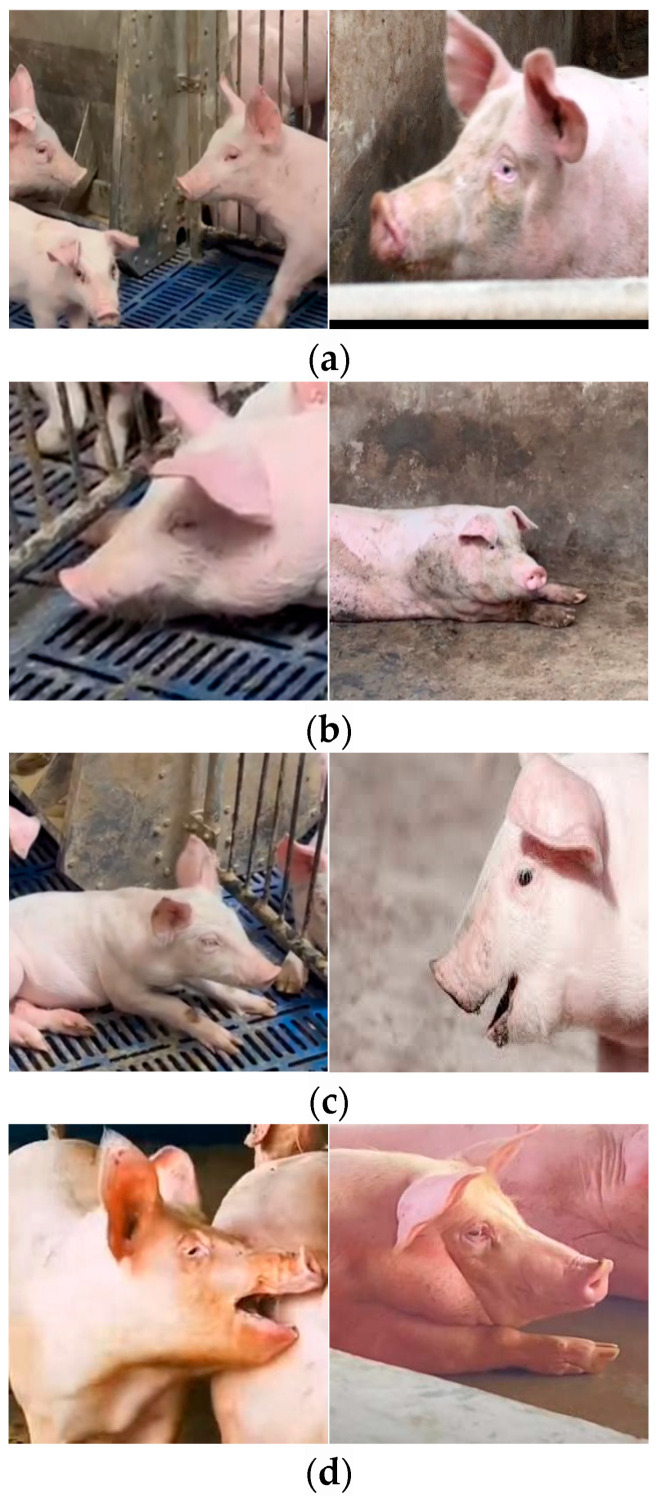
Four representative image categories (**a**) Normal (**b**) Cold (**c**) Cough (**d**) Fever.

**Figure 3 animals-15-03426-f003:**
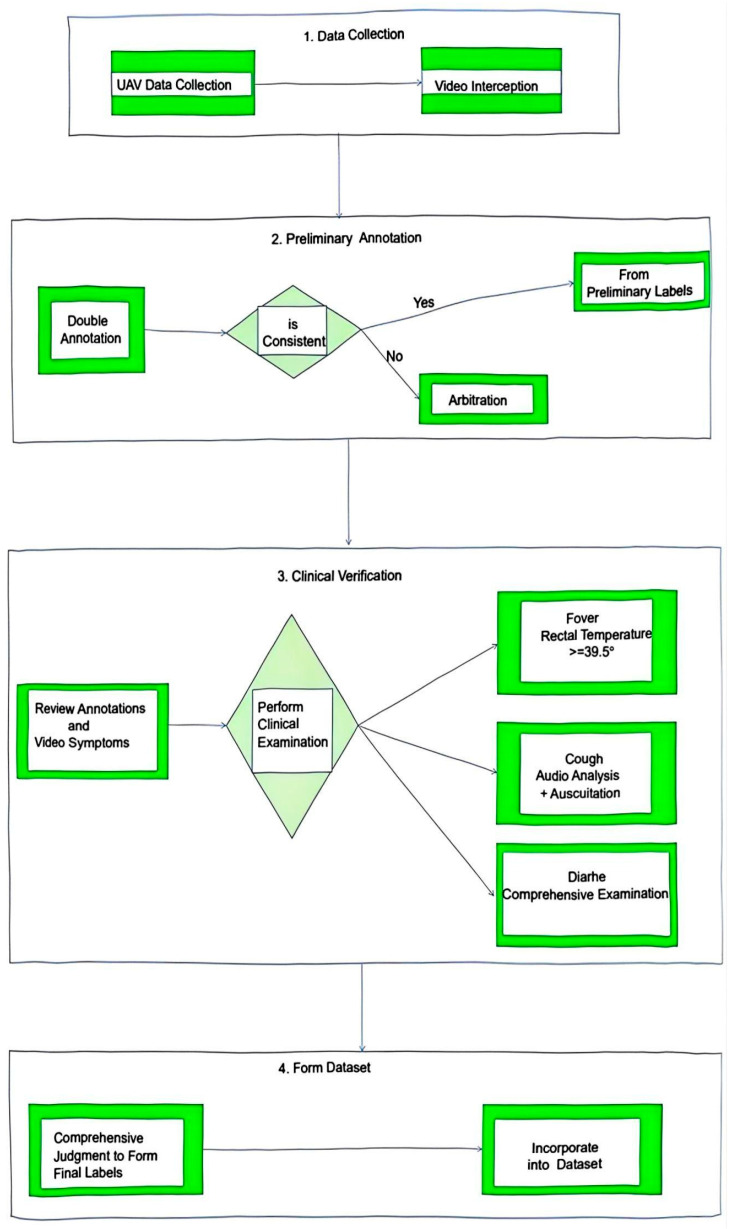
Health status labeling process diagram.

**Figure 4 animals-15-03426-f004:**
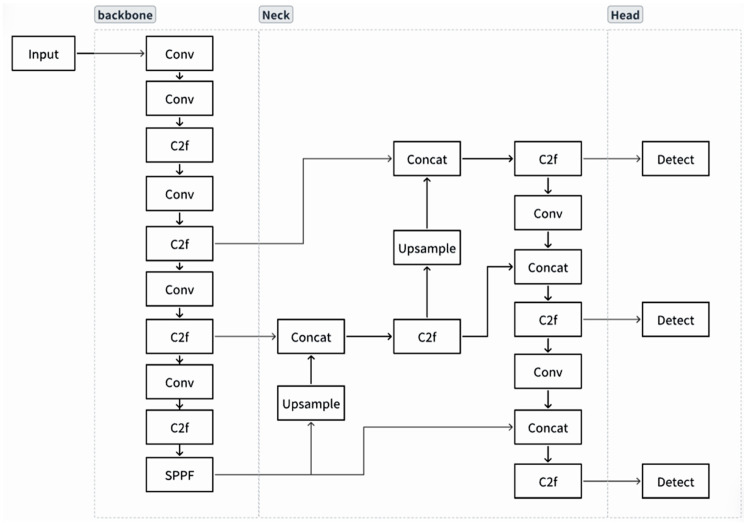
YOLOv8 network architecture: Input, Backbone, Neck, and Head (Prediction).

**Figure 5 animals-15-03426-f005:**
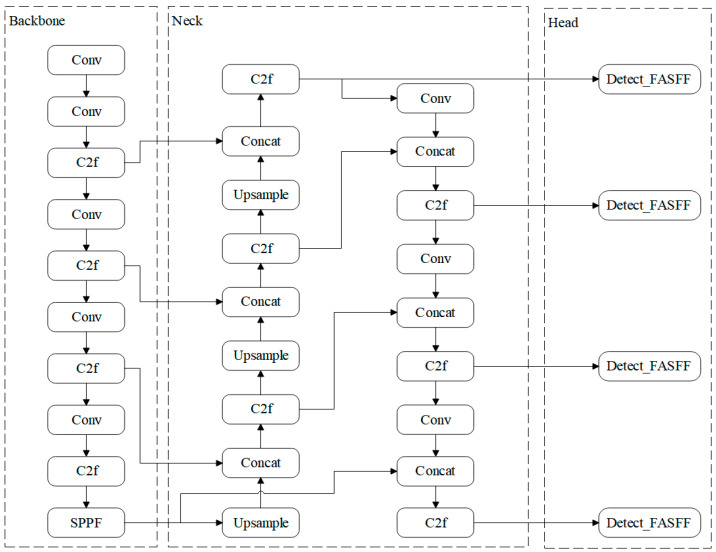
Architectural diagram of the improved FASFF head.

**Figure 6 animals-15-03426-f006:**
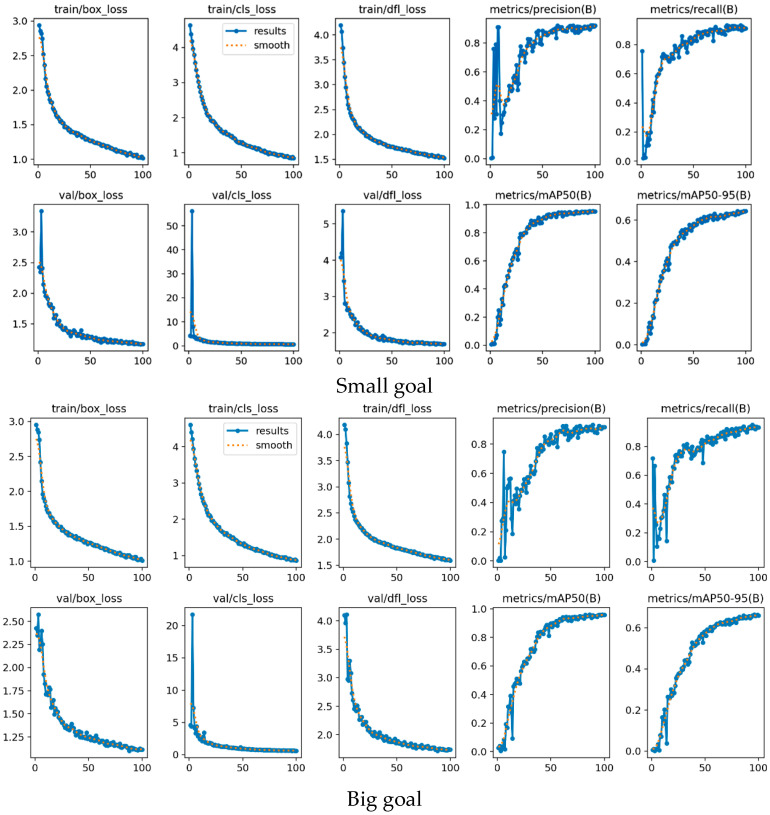
Classification results of the Detect_FASFF-YOLOv8 model.

**Figure 7 animals-15-03426-f007:**
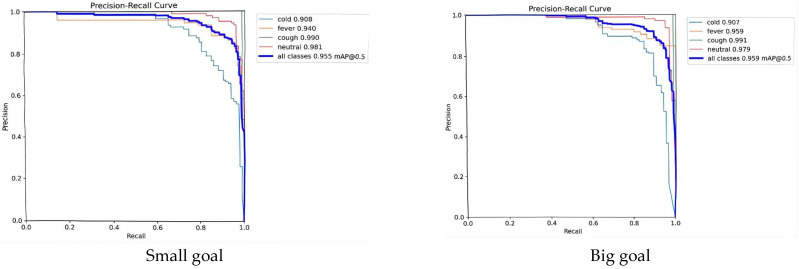
Detection performance of the model, achieving an mAP@0.5 of 0.955 for small objects and 0.959 for large objects.

**Figure 8 animals-15-03426-f008:**
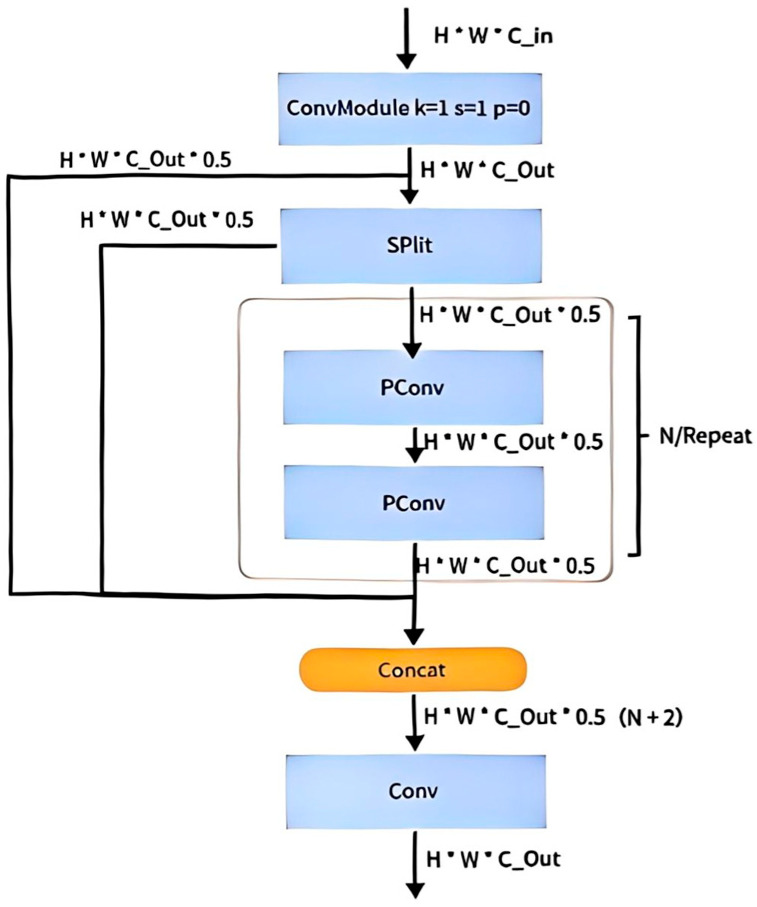
Design of the lightweight PartialConv-YOLOv8 model.

**Figure 9 animals-15-03426-f009:**
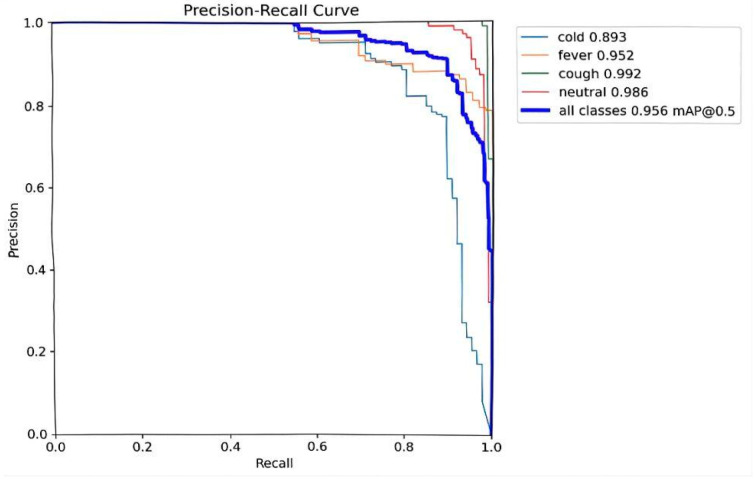
Detection performance of the model, achieving an mAP@0.5 of 0.956.

**Figure 10 animals-15-03426-f010:**
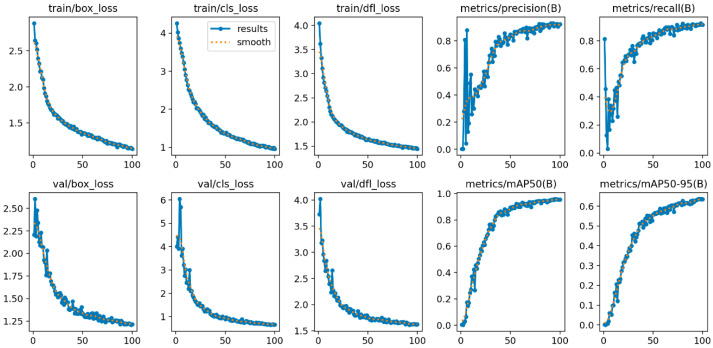
Quantitative results of the lightweight PartialConv_YOLOv8.

**Figure 11 animals-15-03426-f011:**
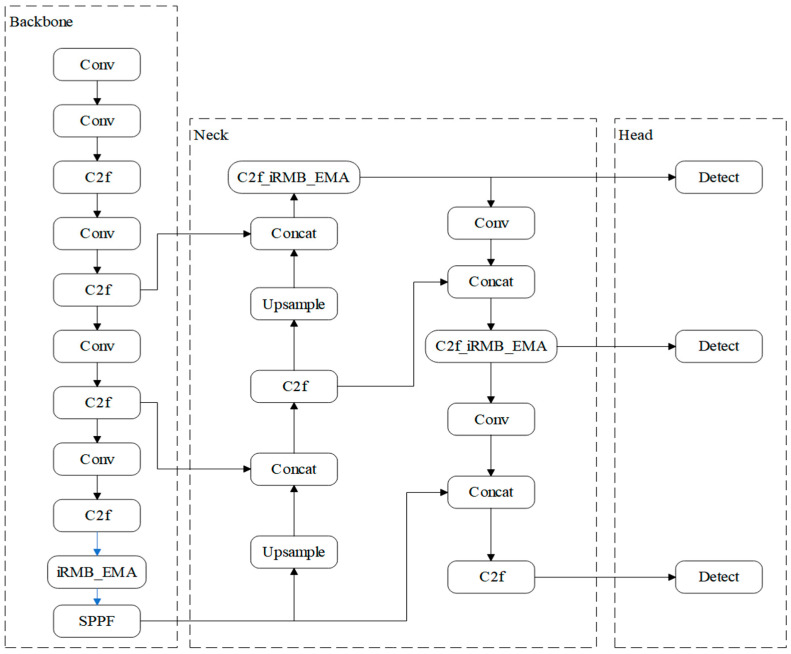
Architecture diagram of the iEMA_Attention Mechanism.

**Figure 12 animals-15-03426-f012:**
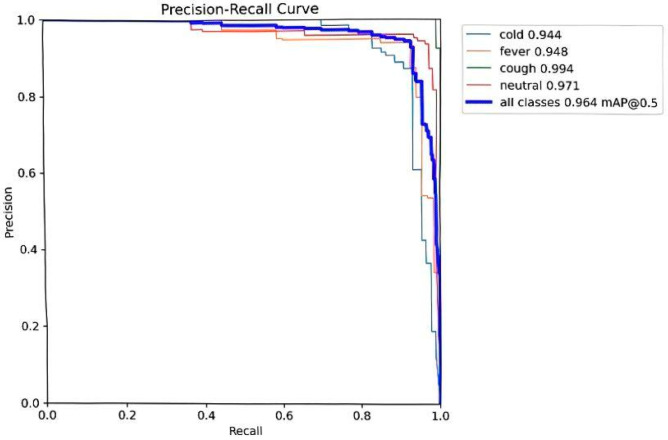
Identification results of the proposed model, showing a mAP@0.5 of 0.964.

**Figure 13 animals-15-03426-f013:**
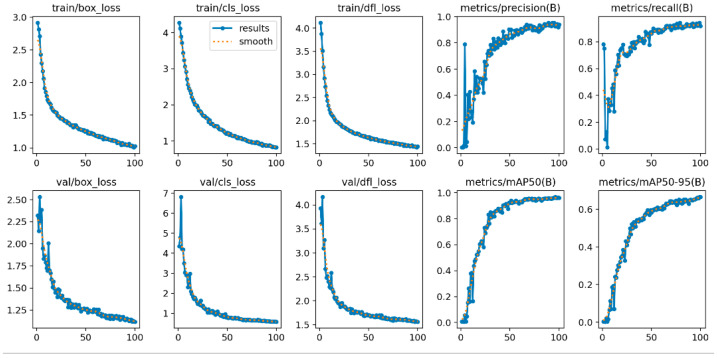
Corresponding recognition results of iEMA Attention mechanism.

**Figure 14 animals-15-03426-f014:**
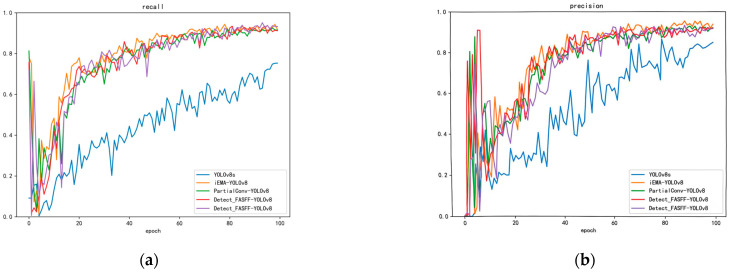
(**a**) Recall (**b**) Precision (**c**) mAP_50_ (**d**) mAP_50–95_. The red Detect_FASFF_YOLOv8 refers to small goal, while the purple Detect_FASFF_YOLOv8 refers to big goal.

**Figure 15 animals-15-03426-f015:**
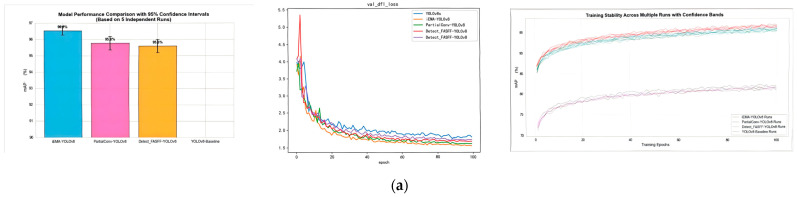
(**a**) Confidence interval with 42 random seeds and loss function (**b**) Confidence interval with 3407 random seeds and loss function (**c)** Confidence interval with 2025 random seeds and loss function. The red Detect_FASFF_YOLOv8 refers to small goal, while the purple Detect_FASFF_YOLOv8 refers to big goal.

**Table 1 animals-15-03426-t001:** Description of the four states.

Status	Nose Condition	Eye Condition	Mouth and Respiration	Overall Expression	Clinical Validation and Objective Measurement
Normal	Moist and glossy	Bright, no discharge	Naturally closed, smooth breathing	Relaxed, alert	Physical examination shows no abnormalities Rectal temperature: 38.5–39.4 °C
Cold	Dry, runny nose	Slight swelling, tear traces	Mouth half-open to aid breathing, nasal congestion	Low-spirited, listless	Veterinarians conduct comprehensive physical examinations Exclude other primary respiratory diseases
Fever	Dry, cracked	Conjunctival congestion, listless	Mouth half-open, gas exhaled hot	Lethargic, unfocused gaze	Rectal temperature ≥ 39.5 °C
Cough	Possible sputum spray	Eyes shut when coughing, restless	Open mouth while coughing, throat constriction	Spasmodic coughing, watery discharge	Audio analysis confirms cough spectrum characteristics Veterinary auscultation reveals abnormal respiratory sounds Clinical examination eliminates environmental stimuli (such as dust)

**Table 2 animals-15-03426-t002:** Dataset partition and per-class sample counts.

Health State	Training Set (n)	Validation Set (n)	Test Set (n)
Normal	950	275	175
Cold	950	275	175
Cough	950	275	175
Fever	950	275	175
Total	3800	1100	700

**Table 3 animals-15-03426-t003:** Ablation study of FASFF components.

Model Variant	Spatial Alignment	Co-Attention	mAP_50_ (%)	FLOPs (G)
Baseline (YOLOv8s)	-	-	82.1	20.6
FASFF w/Naive Fusion	√	-	91.8 (+9.7)	18.1
FASFF w/Alignment Only	√	-	92.5 (+10.4)	18.8
FASFF (Full)	√	√	95.5 (+13.4)/95.9 (+13.8)	15.2/10.2

Footnote: The symbol ‘-‘ denotes that the characteristic is absent; ‘√’ denotes that the characteristic is present.

**Table 4 animals-15-03426-t004:** Comparison of the improved C2f module.

Module	Convolution Type	Parameter Count	Calculation Volume (FLOPs)
Original C2f bottleneck	3 × 3 Standard Conv	C × C × 9	C × C × 9 × H × W
Lightweight C2f bottleneck	3 × 3 Partial Conv	(C/2 × C/2 × 9) × 2	The original calculation quantity 1/2

**Table 5 animals-15-03426-t005:** Ablation study of iEMA-Attention components.

Model Configuration	EMA	iRMB	mAP_50_ (%)	ΔmAP	Params (M)
Baseline + FASFF	-	-	95.5	-	11.2
Variant A	-	√	95.8	+0.3	11.4
Variant B	√	-	96.1	+0.6	11.6
iEMA_Attention (Full)	√	√	96.4	+0.9	11.8

Footnote: The symbol ‘-‘ denotes that the module is lacking; ‘√’ denotes that the module is present.

**Table 6 animals-15-03426-t006:** Benchmarking results of different models.

**Model**		**mAP_50_ (%)**		**mAP_50–95_ (%)**	**FLOPs (G)**		**Recall**	**Box (p)**
Detect_FASFF_YOLOv8 (Small goal) (Big goal)		\ 0.955 0.959		\ 0.643 0.664	\ 15.2 10.2		\ 0.928 0.939	\ 0.916 0.92
YOLOv5s YOLOv8s		0.76 0.821		0.698 0.684	23.8 20.6		0.729 0.753	0.818 0.846
YOLOv11s		0.926		0.682	13.6		0.902	0.918
YOLOv12s YOLOv13s		0.934 0.942		0.679 0.662	7.8 7.1		0.916 0.923	0.927 0.936
Faster R_CNN		0.921		0.65	169.7		0.906	0.845
Rt_Detr		0.888		0.566	103.4		0.833	0.826
PartialConv_YOLOv8		0.956		0.637	6.0		0.916	0.919
YOLOv5s YOLOv8s		0.76 0.821		0.698 0.684	23.8 20.6		0.729 0.753	0.818 0.846
YOLOv11s		0.926		0.682	13.6		0.902	0.918
YOLOv12s YOLOv13s		0.934 0.942		0.679 0.662	7.8 7.1		0.916 0.923	0.927 0.936
Faster R_CNN		0.921		0.65	169.7		0.906	0.845
Rt_Detr		0.888		0.566	103.4		0.833	0.826
iEMA_YOLOv8		0.964		0.665	8.3		0.943	0.92
YOLO 5s YOLOv8s		0.76 0.821		0.698 0.684	23.8 20.6		0.729 0.753	0.818 0.846
YOLOv11s		0.926		0.682	13.6		0.902	0.918
YOLOv12s YOLOv13s		0.934 0.942		0.679 0.662	7.8 7.1		0.916 0.923	0.927 0.936
Faster R_CNN		0.921		0.65	169.7		0.906	0.845
Rt_Detr		0.888		0.566	103.4		0.833	0.826
**Model**	**mAP_50_ (%)**	**mAP_50–95_ (%)**	**FLOPs (G)**	**Recall**	**Box (p)**	**Params (M)**	**Latency (ms)**	**FPS**
**Proposed Models**								
iEMA_YOLOv8 PartialConv_YOLOv8 Detect_FASFF_YOLOv8	0.964 0.956 0.955/0.959	0.665 0.637 0.643/0.664	8.3 6.0 15.2/10.2	0.943 0.916 0.928/0.939	0.92 0.919 0.916/0.92	11.8 6.6 11.2/10.3	12.1 ± 0.8 8.2 ± 0.4 10.5/10.3 ± 0.6	82.6 121.9 95.2/97.1
**Baseline Models**								
YOLOv5s YOLOv8s	0.76 0.821	0.698 0.684	23.8 20.6	0.729 0.753	0.818 0.846	16.1 12.8	20.4 16.7	49.1 59.8
YOLOv11s	0.926	0.682	13.6	0.902	0.918	10.5	10.6	94.3
YOLOv12s YOLOv13s	0.934 0.942	0.679 0.662	7.8 7.1	0.916 0.923	0.927 0.936	6.9 6.5	8.8 7.9	113.6 126.6
Faster R_CNN	0.921	0.65	169.7	0.906	0.845	52.8	46.8	21.5
Rt_Detr	0.888	0.566	103.4	0.833	0.826	31.9	35.5	28.2

**Table 7 animals-15-03426-t007:** Per-class Average Precision (AP) analysis across different architectures.

Model	Normal (AP%)	Cold (AP%)	Cough (AP%)	Fever (AP%)	mAP_50_ (%)
iEMA_YOLOv8	97.1	94.4	99.4	94.8	96.4
PartialConv_YOLOv8	98.6	89.3	99.2	95.2	95.6
Detect_FASFF_YOLOv8	98.1/97.9	90.8/90.7	99.0/99.1	94.0/95.9	0.955/95.9

## Data Availability

The original contributions presented in this study are included in the article. Further inquiries can be directed to the corresponding author.

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
