# Peer review of "Low-Altitude UAV-Based Recognition of Porcine Facial Expressions for Early Health Monitoring"

_animals, 2025, doi:10.3390/ani15233426_

Round 1
Reviewer 1 Report
Comments and Suggestions for Authors
Specific Porcine Expression Recognition via Aerial Inspection
The application is interesting.
This paper’s Detect_FASFF-YOLOv8, PartialConv-YOLOv8, and iEMA-YOLOv8 are based on YoloV8, but there are later newer Yolo models, such as YOLOv12 and YOLOv13.
Include more recent baselines. Check:
Suspicious action recognition in surveillance based on handcrafted and deep learning methods: A survey of the state of the art
Computers and Electrical Engineering 120, 109811
Neural network developments: A detailed survey from static to dynamic models
Computers and Electrical Engineering 120, 109710
Discuss more recent relevant work, such as:
A hierarchical method for locating the interferometric fringes of celestial sources in the visibility data
Research in Astronomy and Astrophysics 24 (3), 035011
Clarify with more technical details about own original contribution.
Clarify with more technical details about advantages of own original contribution.
Proofread to improve the English. Ensure consistent usage of terminology (e.g., always use "YOLOv8" without spaces).
"Error! Reference source not found." appears in the abstract.
Check typos, such as in figure captions (e.g., "reccall" instead of "recall", "percision" instead of "precision").
Explain with more details about loss function.
Table 3 lists multiple models without specifying which is the proposed one in each sub-table.
Explain with more details "IFP-YOLOv8" in Table 3.
Explain with more details about jitter parameters, rotation angles, epochs, batch value, learning rate, et al.
Explain how to avoid bias, such as in the annotation process.
Provide code access link.
Explain with more details on validity and generalizability of results.
Suggest authors to discuss some latest international publications.
Author Response
Dear Reviewer, We sincerely appreciate the valuable feedback from the editor and all reviewers, which has significantly enhanced the manuscript's quality. The comments marked with an asterisk (*) are listed below our responses, with revisions highlighted in red. The page and line numbers are also provided in the replies for your convenience. Below is our response to the feedback:

Reviewer 2 Report
Comments and Suggestions for Authors
Title: Specific Porcine Expression Recognition via Aerial Inspection
Decision: Major revision
Brief summary of the study
The manuscript proposes three YOLOv8-based variants (Detect_FASFF-YOLOv8, PartialConv-YOLOv8, and iEMA-YOLOv8) for detecting four pig facial “health states” (normal, cold, cough, fever) from UAV-captured images. The study describes UAV data collection in a commercial farm in Shanxi (Sep-Dec 2024), curates a 2,800-image labeled dataset (augmented to 5,600), and reports strong mAP@0.5 scores (0.955-0.964) and efficiency gains against several baselines (YOLOv5/11/12, Faster R-CNN, RT-DETR). However, I think important details about ground-truth labeling (physiological confirmation of “fever/cold/cough”), leakage-proof data splitting (by animal/pen/video), training settings, inference speed, and code/data availability limit replicability and the validity of health-state claims.
Specific comments
- Lines 15-25 (Simple Summary). In my opinion, the claims about early “disease alerts” (cold, fever, cough) require clearer clinical ground truth and a validation protocol beyond visual labels; please consider tempering wording or adding evidence on how labels were confirmed (e.g., thermometer, veterinary exam, PCR where relevant).
- Lines 26-44 (Abstract). I think the Abstract should specify the number of animals/pens and the final dataset size (2,800 images; 5,600 after augmentation) to give readers context; currently these appear later only.
- Line 51. Please consider fixing the broken cross-reference “Error! Reference source not found.” in the Abstract.
Introduction
- Lines 56-67. In my opinion, the introduction would benefit from a short taxonomy of prior health vs identity vs behavior recognition studies; currently citations mix identity/pose/segmentation with health state, but do not clarify which achieved clinically validated health labels. Please consider adding 1-2 sentences clarifying this gap. In my opinion, the Introduction could be further strengthened by citing recent work applying large language models for behavioral data extraction in livestock, such as https://doi.org/10.1016/j.atech.2025.101304.
- Lines 87-93. I suggest the authors avoid mechanistic speculation (e.g., linking dry environments to “brief loss of focus” and “subtle eye rotation”) unless supported by cited veterinary sources; otherwise, rephrase as hypotheses motivating automated detection.
Materials and Methods
- Lines 106-117. In my opinion, the UAV platform and Canon EOS 200D needs description of lens/focal length, flight altitude, camera angles, and minimum face pixel size at capture; please consider adding these for reproducibility.
- Lines 128-141. I suggest the authors quantify UAV noise at pig level (dB, distance) and report any desensitization protocol duration/criteria.
- Lines 149-161. I think the health-state labels (normal, cold, cough, fever) require explicit ground truth: who assigned them, based on what measurements, and with what inter-rater agreement? Were fevers validated by rectal temperature? Were cough labels confirmed by audio/clinical exam? Please consider adding a gold-standard definition table and a labeling flow (video → annotation → vet confirmation).
- Lines 160-161. I suggest the authors provide per-class counts (normal/cold/cough/fever) after filtering; the total of 2,800 images is given but not the distribution.
- Lines 197-219. I suggest the authors clarify the split protocol to avoid leakage: were splits done by animal ID or source video (not by image) to prevent near-duplicate frames across train/val/test? Also please confirm that augmented variants of any image were not allowed to cross splits.
- Lines 203-212. I suggest the authors specify the exact image resolution, normalization range, augmentation parameter ranges (rotation degrees, flip probabilities, color jitter ranges), and whether augmentations were applied on-the-fly or offline.
- Lines 214-216. I think “stratified” is good, but please add exact per-class counts for train/val/test and random seeds used to reproduce the splits.
- Lines 221-236. In my opinion, key training details are missing (optimizer, learning-rate schedule, image size, warm-up, weight decay, EMA usage, and early stopping). Please consider adding these. Also, CUDA 10.2 compatibility with an RTX 4060 (Ada) is atypical; please verify driver/CUDA versions used.
- Lines 231-236. I suggest the authors report total training time, best-epoch selection criterion, and number of random seeds trained (with mean±SD mAP).
- Lines 239-248. I think Figure 3 should be referenced in text with a short caption explaining how the baseline head differs from your modified heads to orient readers.
- Lines 250-318. Please consider providing a concise mathematical or pseudo-code description of FASFF (inputs P3/P4/P5 → fused maps; attention weights; loss) and an ablation isolating each sub-component (e.g., spatial alignment vs co-attention), with impacts on mAP and FLOPs.
- Lines 388-466. I suggest the authors more clearly distinguish EMA for model weight averaging vs Efficient Multi-scale Attention; the name “iEMA” blends distinct concepts. A schematic (Figure 10) is referenced, but please add shapes/tensors and where iRMB sits (backbone/neck/head). Also include an ablation showing EMA-only, iRMB-only, and both.
Results
- Lines 489-498 (Figure 13). I suggest the authors fix typos (“map”, “reccall”, “percision”) and add confidence bands or error bars across multiple runs; single-run curves can be noisy.
- Lines 500-504 (Tables). I think the tables need unification and clarity:
- Please consider adding a single consolidated table with params (M), FLOPs (G), latency (ms), FPS, mAP@0.5, mAP@0.5:0.95, Recall, Precision, and Box AP; specify input size used.
- “Detect_FASFF-YOLOv8 (small goal)/(big goal)” looks like two heads or two scales; please clarify.
- “IFP-YOLOv8” appears in the last table but is never defined in Methods; please define or remove.
- Figures 5-12 (Lines 324-470). I suggest the authors ensure all figures display clearly (confusion matrices, examples for each class) and provide per-class APs (Normal/Cold/Cough/Fever) to reveal class-wise weaknesses (e.g., confusion Cold vs Fever).
Discussion
- Lines 525-533. In my opinion, the discussion claims the system “accurately identifies individuals with a body temperature exceeding 39.0 °C,” but no temperature sensing appears in Methods; please consider removing or reframing this, or add methods that measured temperature and how it was linked to images.
- Lines 524-541. I suggest the authors move farm operations claims (coverage %, detection times, LoRa telemetry, 7-day backup, standard NY/T 3254-2023) into a separate case study subsection with methods and evidence, or tone down if anecdotal. As written, they read as operational claims unsupported by the earlier methods.
- Lines 541-566. I think the comparison narrative (YOLOv8 vs Faster R-CNN/RT-DETR/YOLOv11/12) should align with your actual experimental protocol (from-scratch training); please state limitations of that choice and provide pretrained comparisons.
Conclusions
- Lines 603-608. In my opinion, the stated baseline “YOLOv8s mAP@0.5 of 80.1%” is not reported earlier in Results; please include baseline rows in the main table.
- Lines 589-602. Please consider moderating “health monitoring” conclusions unless gold-standard validation is supplied (see earlier comments); alternatively, add a short “clinical validation” plan to future work.
References and formatting
- Lines 635-683. I suggest the authors check reference relevance and formatting (e.g., accessibility notes “accessed 2025-09-20/21,” and missing DOIs.
Language, structure, and minor edits
- Throughout. I think several language issues should be corrected (spacing, punctuation, capitalization: e.g., “Detect_FASFF,” “small goal/big goal,” “map/reccall/percision”). Please consider a careful copy-edit.
- Figure/Table cross-refs. I suggest the authors ensure all figures/tables are referenced in order and that their captions fully explain content (axes, units, thresholds).
Author Response

(The authors gave the same response as above.)

Round 2
Reviewer 1 Report
Comments and Suggestions for Authors
The new version is improved.
Reviewer 2 Report
Comments and Suggestions for Authors
The updated version resolves the concerns previously highlighted.
I have no further comments. In my opinion, the manuscript is now suitable for publication.